# Redefining Hypo- and Hyper-Responding Phenotypes of CFTR Mutants for Understanding and Therapy

**DOI:** 10.3390/ijms232315170

**Published:** 2022-12-02

**Authors:** Tamara Hillenaar, Jeffrey Beekman, Peter van der Sluijs, Ineke Braakman

**Affiliations:** 1Cellular Protein Chemistry, Bijvoet Centre for Biomolecular Research, Science for Life, Faculty of Science, Utrecht University, 3584 CS Utrecht, The Netherlands; m.j.h.hillenaar@uu.nl (T.H.); p.vandersluijs1@uu.nl (P.v.d.S.); 2Department of Pediatric Pulmonology, Wilhelmina Children’s Hospital, University Medical Center Utrecht, Utrecht University, Member of ERN-LUNG, 3584 EA Utrecht, The Netherlands; j.beekman@umcutrecht.nl; 3Regenerative Medicine Center Utrecht, University Medical Center Utrecht, Utrecht University, 3584 CT Utrecht, The Netherlands; 4Centre for Living Technologies, Alliance TU/e, WUR, UU, UMC Utrecht, 3584 CB Utrecht, The Netherlands

**Keywords:** CFTR, protein folding, modulators, hypo-responding mutants, hyper-responders

## Abstract

Mutations in CFTR cause misfolding and decreased or absent ion-channel function, resulting in the disease Cystic Fibrosis. Fortunately, a triple-modulator combination therapy (Trikafta) has been FDA-approved for 178 mutations, including all patients who have F508del on one allele. That so many CFTR mutants respond well to modulators developed for a single mutation is due to the nature of the folding process of this multidomain protein. We have addressed the question ‘What characterizes the exceptions: the mutants that functionally respond either not or extremely well’. A functional response is the product of the number of CFTR molecules on the cell surface, open probability, and conductivity of the CFTR chloride channel. By combining biosynthetic radiolabeling with protease-susceptibility assays, we have followed CF-causing mutants during the early and late stages of folding in the presence and absence of modulators. Most CFTR mutants showed typical biochemical responses for each modulator, such as a TMD1 conformational change or an increase in (cell-surface) stability, regardless of a functional response. These modulators thus should still be considered for hypo-responder genotypes. Understanding both biochemical and functional phenotypes of outlier mutations will boost our insights into CFTR folding and misfolding, and lead to improved therapeutic strategies.

## 1. Introduction

Cystic Fibrosis is an autosomal recessive disease caused by mutations in the Cystic Fibrosis Transmembrane conductance Regulator (CFTR). CFTR belongs to the ATP-binding cassette (ABC) transporter family and functions as an anion channel that mediates the passive but regulated transport of chloride and bicarbonate ions across the plasma membrane at the cell surface. CFTR consists of two transmembrane domains (TMD1 and TMD2) that form the ion-conductance pathway and two nucleotide-binding domains (NBD1 and NBD2) that bind and hydrolyze ATP. The R-domain is unique for CFTR and keeps the two NBDs apart when the channel is closed. The molecular structure of CFTR has been determined in the closed, unphosphorylated, ATP-free state, and the phosphorylated, ATP-bound state [1,2]. Upon phosphorylation, the R-domain repositions, allowing the NBDs to dimerize and ions to pass through the channel [3,4]. To date, the open, ion-conducting structure of the channel has not been determined.

More than 2000 mutations have been identified in the CFTR gene [5], of which >400 have been established to be CF-causing [6]. Disease-causing mutations have been sorted into seven classes based on the nature of the defect: (1) defective protein synthesis, (2) impaired protein trafficking, (3) defective regulation of CFTR channel activity, (4) impaired ion conductance, (5) reduction in CFTR protein levels, (6) decreased protein stability, and (7) decreased mRNA levels [7].

CFTR modulators are available that revert the disease-causing effects of CFTR mutations, theoretically in >90% of CF patients. This is an amazing number because individual modulators have been developed for single mutations. Small molecules that increase the number of CFTR molecules at the cell surface are called correctors. VX-809, for example, was developed for F508del CFTR, which does not fold properly and is degraded [8]. Modulators that enhance channel opening and increase chloride flux are called potentiators. VX-770, for example, was developed for G551D CFTR, which reaches wild-type levels at the cell surface but cannot open its channel [9].

In the past decade, functional classification of CFTR mutations was thought to be beneficial for precision medicine, because modulators for one genotype were thought to be beneficial for all mutants in that class. Correctors should work for classes 2, 5, 6, and perhaps 7 mutants, whereas potentiators should work for classes 3 and 4 mutants. Many if not most mutations display pleiotropic defects, however, and fit into multiple mutation classes. This led to the proposal of an alternative classification system including the original classes plus all possible combinations [10]. Yet, this detailed classification has not improved predictions of therapeutic responses. Despite the co-classification of F508del, G85E, and L206W CFTR variants, patients exhibit highly variable responses to therapy [11]. Even without treatment, patients with identical mutations, F508del homozygotes, showed large clinical variation and recapitulated in vitro in intestinal organoids [12].

Currently, one potentiator, VX-770 (ivacaftor), and three corrector compounds, VX-809 (lumacaftor), VX-661 (tezacaftor), and VX-445 (elexacaftor) are in clinical use and more are in development. Combination therapies are used to rescue different defects with one pill: CFTR biogenesis—domain folding, assembly, and transport—and channel function [13]. Corrector compounds to date have been divided into three categories, based on the additive or synergistic modes of action between categories. Type-I correctors such as VX-809 and VX-661 stabilize TMD1 [14,15,16]. These correctors bind in a hydrophobic pocket in TMD1, stabilizing four helices that are thermodynamically unstable [17]. TMD1 stabilization supports assembly with and of the other domains [15,18]. Type-II correctors such as C4 target NBD2 and its interfaces with NBD1 and TMD2 [19]. VX-445, the most-recently FDA-approved corrector compound, is proposed to act allosterically as a type-III corrector on domain interfaces complementary to the type-I and type-II correctors [20]. Trikafta, a combination of VX-770, VX-661, and VX-445, has been FDA-approved for over 30 CF genotypes, the majority “based on in vitro data in FRT cells indicating that elexacaftor/tezacaftor/ivacaftor increases chloride transport to at least 10% of normal over baseline” [21].

The determining parameter for health versus disease is overall chloride flux through the CFTR channel, which has been measured in patient intestinal organoids for many genotypes and patients. Mutants that lack a detectable functional response are classified as *hypo-responsive* whereas mutants that respond better than average are called *hyper-responsive*. Regardless of the molecular defect, a functional response depends on three distinct parameters: the number of CFTR channels on the cell surface, the probability of the CFTR chloride channel opening, and the current flowing through individual channels. Because functional studies in cells and tissues measure the product of these three parameters, they do not elucidate what is required to rescue the function of a particular mutant.

This study aimed to redefine CFTR hypo- and hyper-responses to clinical modulators. We, therefore, have assessed whether and how modulators affect biosynthesis and cell-surface expression of hypo-responding CF-mutants and how this response compares to hyper-responsive mutants.

## 2. Results

### 2.1. Hypo-Responder Mutants Are Located at the Cell Surface

To determine why certain CFTR mutants lack a functional response to modulators, we first identified whether mutant CFTR had reached the cell surface. We focused on CFTR mutations located in different domains of CFTR (A46D, G85E, R560S, G628R, and L1335P; Figure 1A) that are hypo-responsive to VX-809 and VX-770 in patient intestinal organoids [22], (see also Section 4, Appendix A). To eliminate the effects of genetic background, we used a laboratory cell line for this study. The cell-surface population of CFTR was determined by treating HEK293T cells expressing wild-type and mutant CFTR with membrane-impermeant Sulfo-NHS-SS-Biotin. Biotinylated proteins were then pulled down with Neutravidin beads from detergent lysates of these cells, and detected by Western blotting using antibodies against CFTR. Cell-surface levels were compared with total CFTR levels in the cell (input), with actin as control (Figure 1B and Appendix A).

In the endoplasmic reticulum (ER), wild-type CFTR is core glycosylated at two sites in TMD2. When CFTR migrates from the ER to the Golgi complex, oligo-mannose-type glycans are modified to complex-type glycans, which results in a shift in electrophoretic mobility of CFTR from the 150-kDa ER form called ER (or B band) to the 180-kDa Golgi-modified form called G (or C band) (Figure 1B, input) [23,24]. Eighty percent of wild-type CFTR molecules were complex glycosylated and hence had left the ER and resided in or beyond the Golgi compartment.

The fate of every (mutant) CFTR molecule is either transport to the Golgi compartment (and onwards to the cell surface), or degradation. Without modulators, the export of G85E and R560S CFTR mutants from the ER was undetectable against the smear of aggregates, whereas small quantities of A46D and G628R left the ER and appeared as fuzzy, weak Golgi bands (Figure 1B, input). The overall expression of these four mutants was much lower than that of wild-type CFTR, suggesting massive protein degradation. L1335P CFTR did better, with 40% of wild-type CFTR input levels, and in steady state ~35% in the Golgi form. This mutant was unusual in the relatively high fraction of protein in the ER, implying less degradation of the ER-resident form than for wild type and other mutants.

The cell-surface pool of wild-type CFTR consisted of the complex glycosylated form, suggesting that only the Golgi-modified version of CFTR was transported to the plasma membrane (Figure 1B). A small quantity of ER form may appear on the plasma membrane as well, as the CFTR-band smeared all the way down to the ER-form mobility. Because the abundant protein actin was not detectable in the cell-surface pool, we concluded that none of the reactive biotin had entered the cell and labeled intracellular CFTR during the surface biotinylation. This implied that a small quantity of ER band on the cell surface may well be real. The mostly ER-retained CFTR mutants A46D, G85E, R560S, and G628R were virtually absent from the cell surface (<10% of wild-type CFTR cell-surface levels), with again a trace of ER form. As expected, L1335P did reach the cell surface. Surprisingly, surface biotinylation labeled both the ER and Golgi-modified forms of L1335P CFTR. The percentage of ER form was higher than that of wild type for this mutant, as in the input, suggesting that the cell recognized this mutant as different from the wild type but also distinct from misfolded CFTR mutants that are degraded effectively.

None of the tested CFTR mutants reached cell-surface levels comparable to wild-type CFTR. The low number of CFTR molecules at the cell surface could explain why these mutants are not sufficiently functional and cause the CF-disease phenotype.

### 2.2. VX-809 Increases Mutant CFTR Levels

We next determined whether corrector VX-809 affects the total and cell-surface populations of A46D, G85E, R560S, G628R, and L1335P CFTR. To this end, transfected cells were pre-incubated with 3 µM VX-809 for 18 h before surface biotinylation (Figure 1B, VX-809 lanes). We used G85E as negative control because VX-809 does not affect total levels of G85E and has not responded at all to VX-809 in our assays [25,26,27,28,29]. Indeed, VX-809 did not increase the total number of G85E CFTR molecules nor G85E cell-surface levels (Figure 1B).

The cell-surface and total levels of wild-type CFTR and the R560S, G628R, and L1335P mutants increased ~1.5–2 fold upon VX-809 addition, implying that these functionally hypo-responding mutants were responsive to VX-809. A46D even showed a 3–4-fold increase at the cell surface due to VX-809, to ~35% of wild-type CFTR. In contrast, R560S and G628R only reached ~15% of wild-type CFTR. Since the number of L1335P CFTR molecules in the cell was already higher in the control condition, the 2-fold increase led to protein levels of ~70% of wild-type CFTR. Both CFTR forms of L1335P were detected at the cell surface and VX-809 did not change the percentage of Golgi-modified CFTR. We concluded that the A46D, R560S, G628R, and L1335P CFTR mutants were responsive to VX-809, as the total amount of CFTR increased. Only A46D responded with an increase in %Golgi, both at the cell surface and in the cell, implying that the A46D-CFTR pool that escaped degradation was transported faster to the Golgi complex and cell surface or was more stable there.

The acute addition of VX-770 (for 5 min) did not significantly change the total amount of CFTR in the cell or at the cell surface. This was expected because the time of VX-770 incubation (15 min) was too short to greatly change biosynthesis or degradation. As VX-770 did not affect CFTR levels, combination treatment showed CFTR levels comparable to the VX-809 condition.

We have now established that the hypo-responder mutants did improve folding and/or stability upon VX-809 treatment, with A46D responding strongly even. The question is whether their correction was insufficient for function or whether they have unusual folding defects.

### 2.3. Hypo-Responder CFTR Mutants Have Mutation-Specific Molecular Defects

We, therefore, determined the folding defects in A46D, G85E, R560S, G628R, and L1335P, which must contribute to their causing disease. Cells expressing (mutant) CFTR were radiolabeled for 15 min and chased for 2 h. Detergent cell lysates were subjected to limited proteolysis followed by immunoprecipitation with CFTR domain-specific antibodies.

Figure 2A (top panel) shows core-glycosylated CFTR directly after radiolabeling (150 kDa) before proteolysis. At the end of the pulse labeling, the levels of radiolabeled wild-type and mutant CFTR were similar, suggesting that the mutations did not cause degradation during synthesis. Limited proteolysis was used to probe CFTR conformation. This approach allows unbiased analysis of changes in CFTR conformation with time under various conditions. Limited proteinase-K digestion of mutant CFTR leads to changed digestion patterns on SDS-PA gels because misfolded proteins and folding intermediates are cleaved more efficiently than well-folded, more compact proteins [18,24,30,31,32,33,34].

After limited proteolysis of newly synthesized wild-type CFTR, domain-specific protease-resistant fragments were immunoprecipitated with antibodies against NBD1, TMD1, TMD2, or NBD2. We recently identified the boundaries of the domain-specific proteolytic fragments at early (0 h) and late (2 h) chase times [18,31]. This information was used to uncover small changes in CFTR conformation and the folding profile of the CFTR mutants. Immediately after synthesis (0 h chase), three out of four wild-type CFTR domains are folded already and yield prominent proteolytic fragments: one for NBD1 (N1a), three for TMD1 (T1a-c), and one for TMD2 (T2b); NBD2 is not folded yet and yields a barely detectable fragment (N2a) (Figure 2A, wt) [18,31]. The presence and quantity of these fragments are a measure of co-translational CFTR domain folding.

During the 2 h chase, wild-type CFTR continues folding and is transported to the Golgi complex (Figure 2B top panel, Appendix A) [18,31]. Post-translational assembly of the domains leads to increased protease resistance, resulting in larger CFTR fragments: three for TMD1 (T1d-f), one for TMD2 (T2c), and one for NBD2 (N2a), now a strong band (Figure 2B other panels, Appendix A) [18,31]. These late, larger fragments appear simultaneously due to the assembly of the two TMDs with each other and with NBD2 [18,25,35].

To determine the molecular defects of the hypo-responder mutants, we analyzed the proteolytic fragments of the radiolabeled CFTR mutants. All mutants deviated from wild type and each other, except for the similarity of the two NBD1 mutants, R560S and G628R.

G85E is a good negative control because it was negative in biochemical assays [25,26,27,28,29] and had been well-established to lack significant rescue to wild-type by Trikafta [25]. The G85E mutation is located in TMD1 and is important for the helical packing of this domain. The mutated TM1 does not integrate properly into the membrane [36,37], due to the destabilizing effect of the negatively charged glutamate (E) in the membrane. The destabilization of G85E TMD1 is visible by a mild electrophoretic downshift in TMD1 fragment T1a (Figure 2A and Appendix A, [25]). As expected, G85E prevented assembly of the CFTR domains: the T1d-f, T2c, and N2a fragments were absent after 2 h of chase (Figure 2B and Appendix A). Consistent with steady-state data (Figure 1B), not only mutant G85E but also R560S and G628R were retained in the ER and degraded during the 2 h chase, whereas A46D and L1335P escaped to the Golgi complex and cell surface to some extent (Figure 2B and Appendix A).

The R560S and G628R mutations are located in NBD1 and disturbed the folding of this domain because NBD1 did not acquire protease resistance and the 25-kDa N1a fragment was absent (Figure 2A and Appendix A). TMD1 and TMD2 folding of R560S and G628R were similar to that of wild-type CFTR, meaning that the primary folding defect is limited to NBD1 and does not transmit immediately to another domain. As in the case of G85E and F508del CFTR [18,25,30,38], the primary defect did spread during the chase, because both NBD1 mutants also lacked the T1d-f, T2c, and N2a fragments, demonstrating that CFTR domain assembly was impaired by each mutation. As a consequence, the NBD1 mutants never left the ER and were degraded instead (Figure 2B and Appendix A). The steady-state data (Figure 1B) demonstrate that, nonetheless, a small quantity must have escaped and reached the cell surface.

The TMD1 domain of newly synthesized A46D yielded a fragment pattern that seems to have shifted up, with larger ‘T1a’ and ‘T1c’ fragments (Figure 2A,C and Appendix A). A change in the size of a fragment in mutant CFTR can have multiple causes. Conversion of alanine 46 to the negatively charged aspartic acid cannot have caused a change in electrophoretic mobilities of wild-type T1a nor T1c, because residue 46 is not part of either of these wild-type fragments: T1a consists of residues S50-S258 and T1c of residues S50-L383 [18,25]. The A46D mutation must have protected the L49 cleavage site leading to larger fragments, most likely because it elicited a conformational change in the TMD1 N-terminus. The A46D ‘T1a’ fragment that replaced T1a may well be D36-S258, and wild-type T1c then probably would have changed into A46D T1d, D36-L383, except that this band ran higher than wild-type T1d, which is more easily seen in Appendix A.

Two hours of chase did not change the early A46D-CFTR fragment pattern much except that the weak A46D ‘T1d’ turned into a stronger ‘T1d-f’. T1d-f all contain the A46D mutation (Figure 2B,C and Appendix A). As wild-type T1f contains the complete N-terminus of TMD1 [18], the A46D-induced mobility decrease in T1f suggests that the mutation indeed affected the mobility of T1d as well. The electrophoretic mobility changes of the early A46D TMD1 fragments hence were caused by the protection of L49 (with the shifted T1a indeed being D36-S258 and T1c changing into ‘T1d’), whereas the mobility changes of the late A46D TMD1 fragments ‘T1d-f’ were caused by the introduction of a charge in these fragments rather than a changed cleavage site (Figure 2B,C and Appendix A) [18]. In this scenario, A46D is unlikely to affect the protease sensitivity of the C-terminal half of TMD1. Immediately after synthesis, L49 must have been protected already in the A46D-induced conformation, in contrast to the wild-type conformation. During the chase period, A46D-CFTR domain assembly further increased the protection of the Lasso motif [3,18]. Indeed, a small fraction of A46D CFTR reached the Golgi complex (Figure 2B and Appendix A) and cell surface (Figure 1B).

The lack of CFTR function of A46D, G85E, R560S, and G628R can be explained by both an insufficient number of CFTR molecules at the cell surface (Figure 1B) and by multiple folding defects in each mutant protein, as detected by the proteolysis folding assay (Figure 2A,B) [18]. The L1335P mutant was different, as it was located at the cell surface in the control condition, albeit not to wild-type levels. Immediately after synthesis (at 0 h chase), L1335P CFTR fragmented like the NBD1 mutants R560S and G628R, except that NBD1 was folded like wild-type NBD1 and yielded the protease-resistant N1a fragment. After the 2 h chase, only a small fraction of L1335P had reached the Golgi complex, and we again detected more L1335P inside the ER. Despite some export from the ER after 2 h of chase, domain assembly turned out to be impaired and deviant from any mutant examined so far. Whereas TMD1 showed the expected quantity of T1d-f, domain-assembly fragments of TMD2 (T2c) and NBD2 (N2a) were completely absent. A smaller NBD2 fragment was barely detectable (Appendix A). This suggests that L1335P disrupts the folding of NBD2 and its assembly with TMD2 without affecting the N-terminal half of CFTR (TMD1-NBD1). We concluded that incomplete domain assembly contributed to the dysfunction of L1335P CFTR channel activity.

### 2.4. VX-809 and VX-770 Cause Discrete Conformational Responses in Wild-Type CFTR

Corrector VX-809 increased CFTR levels of all hypo-responder mutants except G85E (Figure 1). For most mutants, however, this increase did not lead to wild-type CFTR levels. The insufficient cell-surface levels of A46D, R560S and G628R CFTR molecules combined with multiple conformational defects is the likely cause of their dysfunction. L1335P CFTR cell-surface levels were closest to wild type (70%), which might have been sufficient for a functional response. Since this mutant is not functional, we envisage the conformational difference, and thereby inability to open and close the chloride channel, to be CF-causing.

We next asked whether the conformation of CFTR mutants was rescued towards wild-type levels. As for Figure 2, CFTR was radiolabeled for 15 min and chased for 2 h, but now both in the presence of VX-809 (3 μM). VX-809 binds TMD1 and increases the ratio of TMD1 fragments T1aa and T1a (Figure 3A–C; [25]). Both T1a and T1aa fragments are cleaved at the same C-terminal residue. The difference between the fragments lies in their N-terminal boundaries, which is residue 36 for T1aa (as for T1d) and residue 50 for T1a [18,25]. The increase in T1aa/T1a ratio is caused by improved packing of ICL1 and the N- and C-termini of TMD1, leading to increased protease protection of L49. A consequence of improved TMD1 is the increase in domain assembly, seen as fragments T1d-f from TMD1, T2c from TMD2, and N2a from NBD2. In summary: corrector compound VX-809 binds and improves TMD1 during synthesis (Figure 3A–C), which then leads to improved domain assembly (Figure 3A), and this stabilizes full-length CFTR, which is recognized by cellular factors and leads to an ~two-fold increase in wild-type CFTR levels (Figure 3A, top panel). We used this information to identify whether and where hypo-responder mutants deviated from wild-type CFTR in their conformational response to VX-809.

We similarly examined the effect of clinical potentiator VX-770 on wild-type CFTR (Figure 3A), because a potentiator causes conformational changes as well, but then acutely and not necessarily during synthesis. Because of this, and as reported before, the stability of CFTR indeed may be affected by chronic exposure to VX-770 [30,39,40]. In the pulse-chase assays, however, total protein levels and transport of wild-type CFTR to the Golgi complex did not change when VX-770 was added chronically (Figure 3A; [30]). VX-770 did change CFTR conformation but only after domain assembly (2 h chase). The potentiator increases the amount of T1d (D36-L383) at the expense of T1f (M1-L383) (Figure 3A,D,E; [30]), indicating that it causes deprotection—i.e., more exposure to protease—of the N-terminal part of the Lasso motif. Another effect of VX-770 is the deprotection of the TMD2 fragment T2c (Figure 3A; [30]). The changes in the proteolytic pattern of CFTR suggest that the assembly of CFTR domains becomes less tight upon VX-770 binding. The G551D CFTR mutant for which VX-770 was developed indeed needs this loosening, as the mutant is a highly stable protein but a closed channel [41].

In summary, the primary and secondary effects of VX-809 and VX-770 on wild-type CFTR conformation are detectable by limited proteolysis. We used this information to determine whether and how hypo-responding CFTR mutants respond to these modulators.

### 2.5. Hypo-Responder CFTR Variants Show VX-809-Induced Proteolytic Responses

We analyzed whether the functional-hypo-responder variants A46D, G85E, R560S, G628R, and L1335P showed the VX-809-specific proteolytic patterns seen in Figure 3 for wild-type CFTR. The mutants were expressed in HEK293T cells, radiolabeled for 15 min, and chased for 0 or 2 h in the absence or presence of VX-809 (Figure 4 and Appendix A). Detergent cell lysates were subjected to limited proteolysis. Immediately after the pulse, the T1aa/T1a ratio is the primary indicator of a VX-809 response, caused by protection of the N-terminus of T1a around residue L49. G85E is not sensitive to VX-809, has a down-shifted ‘T1a’, and does not yield a T1aa-like band (Figure 4A,B; [25]). As expected, R560S, G628R, and L1335P responded like wild-type CFTR, with a doubling of the T1aa/T1a ratio (Figure 4A).

A46D CFTR was proteolyzed into larger TMD1 fragments than wild-type CFTR, due to the shielding of L49, coincidentally the same region that VX-809 protects. It is no surprise, then, that VX-809 did not change the TMD1 fragment pattern in the A46D mutant. Yet, VX-809 must have interacted with A46D TMD1, as it caused a strong increase in A46D CFTR domain-assembly fragments T2c from TMD2, N2a from NBD2, and upshifted T1d-f from TMD1 (likely T1e and T1f as reasoned above). This implied a strong response of A46D CFTR conformation to VX-809.

VX-809 did not rescue the primary folding defect of R560S and G628R CFTR—their lacking NBD1 folding. Protection of the N-terminus by VX-809 nevertheless led to improvements in domain assembly for these mutants—increased T1d–f, T2c, and N2a fragment intensities—and increased ER-to-Golgi transport (Figure 4C and Appendix A). G628R conformation improved more by VX-809 than R560S CFTR but still did not reach wild-type-like fragment intensities upon VX-809 addition. We concluded that the R560S mutation appeared more disruptive for CFTR domain assembly than G628R. The primary defect of L1335P in NBD2 was not rescued either. VX-809 improved L1335P CFTR assembly of the TMD1 (fragments T1d–f) and NBD1 domains during the chase but did not show any sign of domain assembly of TMD2 nor NBD2 (lack of T2c and N2a, with the consistent appearance of a smaller NBD2-derived, 596-epitope-containing fragment).

We concluded that the three hypo-responder NBD mutants showed the VX-809-induced proteolytic pattern in TMD1 and that all mutants except G85E had improved by VX-809, in CFTR quantity and/or transport to the Golgi complex. The primary folding defects caused by these mutations were not rescued by VX-809, despite improved cell-surface levels and conformations, which may contribute to their lack of functional response upon rescue. Complete rescue of mutant CFTR may well require the correction of the primary folding defect.

### 2.6. VX-770 Induces Conformational Responses in Hypo-Responder Mutants

To determine whether the hypo-responder mutants are sensitive to VX-770 as well, we analyzed the effect of the potentiator on the conformation of the mutants. As for wild-type CFTR, VX-770 had no detectable effect on the mutants immediately after synthesis (Figure 4B, top panel). After 2 h of chase, G85E, R560S, and G628R still did not show a detectable effect of VX-770, most likely because these mutants had no or scarcely assembled domains (Figure 4C and Appendix A). Similar to wild-type CFTR, VX-770 had no effect on NBD1 nor NBD2, it destabilized T2c and increased the upshifted T1d fragment of A46D CFTR, but without a decrease in other TMD1 fragments. L1335P responded with a strong increase in T1d at the expense of T1a, T1c, and T1f, implying deprotection of the far N-terminus of CFTR, up to residue D36, the N-terminus of T1d (Figure 4C). Interestingly, L1335P showed a stronger response to VX-770 than wild-type CFTR. The L1335P mutation is located in the NBD2 domain at the NBD2-ICL2 interface, and it prevents the domain assembly of NBD2 that leads to its proteolytic stability. VX-770 enhanced the packing of the domains, leading to the protection of ICL2. Surprisingly, whereas VX-770 decreased the intensity of T2c in wild-type CFTR, the L1335P mutant lost T2b but gained some T2c, suggesting a subtle conformational changes and a net destabilization of TMD2.

VX-809 induced a distinct, primary conformational change in the hypo-responder mutants R560S and G628R, which led to their improved domain assembly and exit from the ER, albeit insufficient for functionality. A46D and L1335P showed unique responses to VX-809 and were the only mutants that reached sufficient domain assembly for VX-770 to impact their conformation. Also here, their conformational responses were distinct from those of wild-type CFTR, which may well explain their lack of functional response. With these conformationally distinct responses of some hypo-responder mutants, we wondered whether hyper-responding mutants responded simply stronger or also distinctly.

### 2.7. Similar to Hypo-Responders, Some Hyper-Responding Mutants Are Retained in the ER

We, therefore, selected CFTR mutants that respond better than average to VX-809 and VX-770. The hyper-responding mutants were located in the same three domains as the hypo-responders: TMD1 (L206W and I336K), NBD1 (G461R), and NBD2 (G1249R and S1251N) (Figure 5A). The (most common) F508del mutation (in NBD1) was included because the clinical efficacy of modulators is variable in patients carrying F508del mutations, resulting in both hypo- and hyper-responding effects in patients [11,12].

Using cell-surface biotinylation, as in Figure 1, we first assessed whether the mutant proteins were transported to the cell surface. In common with the hypo-responding mutants, F508del, L206W, and I336K were mainly retained in the ER and absent from the cell surface (Figure 5B). None of these CFTR mutants showed more than 10% of wild-type cell-surface levels, suggesting that the low number of CFTR molecules at the cell surface is the cause of, or contributing to, the CF-phenotype.

In contrast, the G461R, G1249R, and S1251N mutants were already present at the cell surface in control conditions. G461R showed 35% of wild-type cell-surface levels. Strikingly, cell-surface levels of G1249R and S1251N were even higher than those of wild-type CFTR (Figure 5B). We concluded that for G461R, G1249R, and S1251N, the CF-phenotype must be caused by an inability to open the chloride channel or defective ion conductance, rather than by an insufficient number of channels at the plasma membrane.

### 2.8. VX-809 Increases Cell-Surface Levels of Hyper-Responding CFTR Mutants

We next determined the effect of VX-809 on the cell-surface levels of the hyper-responding mutants (Figure 5B). The chronic addition of VX-809 increased input CFTR levels for all mutants. Wild-type CFTR and the F508del, G461R, G1249R, and S1251N mutants increased two-to-three-fold. Strikingly, L206W and I336K CFTR levels increased ~four-fold in the presence of VX-809, and these mutants were no longer retained in the ER.

The F508 deletion impairs CFTR expression at the plasma membrane [24,25,42,43]. The two-fold improvement indeed only led to a minor increase in cell-surface expression (~10% of wild-type CFTR levels). Consistent with their strong VX-809-improved expression, L206W and I336K ER-to-Golgi transport was rescued, and cell-surface levels were restored to ~70–90% of wild-type CFTR. The already wild-type-like G461R, G1249R, and S1251N CFTR cell-surface levels further increased in the presence of VX-809, as did those of wild-type CFTR.

As expected, the acute addition of potentiator VX-770 did not affect the total number of CFTR molecules in the cell or at the cell surface significantly, nor did it influence the effect of VX-809 in combination therapy.

### 2.9. Molecular Defects Are Consistent with CFTR Dysfunction

To characterize the folding defects of the hyper-responding CFTR mutants, we used radiolabeling, as described above (Figure 6 and Appendix A). CFTR domain folding is shown in Figure 6B (0 h chase) and domain assembly in Figure 6C (2 h chase). The F508del mutation induces NBD1 misfolding, shown by the absence of N1a (Figure 6B; [24,25]). The misfolding of NBD1 disrupts the assembly of the other domains [18,33], leading to the absence of domain assembly fragments T1d-f, T2c, and N2a (Figure 6C; [30]).

L206W and I336K were mainly retained in the ER, in common with F508del CFTR (Figure 6B top panel, Appendix A). Radiolabeling combined with limited proteolysis was used to identify whether molecular defects could explain the lack of functionality. We did not detect folding defects in the individual domains of L206W and I336K CFTR (Figure 6B), but the later chase time revealed severely impaired domain assembly: decreased or absence of T1d-f, T2c, and N2a fragments (Figure 6C and Appendix A). Defective domain assembly led to impaired ER-to-Golgi transport in the radiolabeled proteins (top panel), all leading to an insufficient number of CFTR molecules at the plasma membrane.

The conformations of G461R, G1249R, and S1251N were close to that of wild-type CFTR at early and late chase times, suggesting that the molecular defects were not causing large conformational changes in the protein. Yet, all three mutants have a functional defect [22,44,45,46], with impaired chloride ion conductance, implying at least subtle conformational defects. Only S1251N has been classified as class 3 [44].

### 2.10. Proteolytic Responses to Modulators Are Like Those of Wild-Type and F508del CFTR

All functionally hyper-responding CFTR mutants showed an increase in total steady-state and cell-surface levels upon VX-809 addition. As expected, all mutants showed the typical effect of VX-809 on the T1aa/T1a ratio, in common with wild type (Figure 6A), with a slightly milder increase for L206W CFTR. As before, the improvement of TMD1 led to enhanced domain assembly, with the larger proteolytic fragments T1d-f, T2c, and N2a after 2 h (Figure 6C and Appendix A). The consequence was improved protein stability and ER-to-Golgi transport for all mutants (Figure 6C, top panel). Similar to the hypo-responding NBD1 mutants, VX-809 did not improve the folding nor protease resistance of the defective F508del NBD1 (Figure 6B,C and Appendix A).

All CFTR mutants showed the proteolytic response of VX-770 on CFTR conformation, only after a chase and domain assembly: a strong decrease in TMD2 fragment T2c and an increase in T1d, either at the expense of larger fragment T1f or of T1a (Figure 6C and Appendix A). The VX-770 effects were stronger in VX-809-improved mutants.

### 2.11. Hypo- and Hyper-Responding CFTR Mutants Respond Similarly to Modulators

The hyper- and most hypo-responding mutants were affected by the modulators with respect to CFTR conformation, the total number of CFTR molecules in the cell, and the cell-surface population. Most of these responded like wild type but some deviated from the typical patterns. To compare the two mutant populations, we plotted quantified data from the experiments, as shown above.

To determine whether the response of radiolabeled hypo-responding mutants and hyper-responding mutants is comparable in the presence of VX-809, the fraction of CFTR that had left the ER after 2 h of chase in the control condition was plotted against the fold change in total CFTR after 2 h of chase caused by VX-809. Most mutants responded to VX-809 in common with wild-type CFTR, with a 1.5–2.5-fold increase in total protein levels (Figure 7A). Striking was the hyper-response of L206W and I336K mutants, for both increasing the total amount of CFTR with 3.5–4-fold in the presence of VX-809 despite their milder response on the N-terminus of TMD1, the T1aa/T1a ratio (Figure 6A).

When plotting the same parameters but from the steady-state instead of kinetic data, the pattern was similar (Figure 7B), except that four out of five hyper-responding and one hypo-responding mutant were ~three-to-four-fold improved by VX-809 in the steady-state conditions. This implies that these cell-surface-localized rescued mutants benefited more from VX-809 stabilization at the cell surface than in the ER.

Figure 7A,B shows roughly two clusters in every graph, the red cluster with hypo-responder mutants, and the green cluster with hyper-responding mutants, with limited mixing between the clusters. This division is solely caused by the low number of hypo-responder CFTR molecules that exited the ER, leading to low residual activity (Figure 7C). All mutants except G85E responded at least like wild-type to the presence of VX-809 and functional hypo-responder mutant A46D turned out to be a biosynthetic hyper-responder. The difference between most hypo- and hyper-responders, therefore, is not caused by the degree of modulator response but predominantly by residual activity of each mutant.

VX-809 not only increased the total amount of (mutant) CFTR protein but also increased ER-to-Golgi transport of most CFTR mutants, seen by the increase in %G in the presence of VX-809 (Figure 1, Figure 4, Figure 5, Figure 6, and Figure 7D). Some mutants displayed a larger total amount of CFTR than expected from their ER export due to a slightly increased quantity in the ER, which is not degraded and not yet transported. Functional hypo-responder L1335 (and to a much lesser extent, the hyper-responders G1249R and S1251N) reside significantly longer in the ER, suggesting an alternative chaperone-mediated triage pathway, perhaps similar to N1303K [47,48].

We concluded that hypo-responders are biochemically responsive to modulators and that some of them may well be classified as hyper-responders, despite their poor functional response.

### 2.12. VX-445 Rescues F508del CFTR Constructs through Interdomain Contacts

To identify whether the VX-809 response of hypo-responder mutants can contribute to functional rescue of the mutants, we combined VX-809 with the second-generation corrector VX-445. VX-445 synergizes with type-I and -II correctors and, therefore, was suggested to act as a type-III corrector, stabilizing the NBD1 domain [49]. The binding site of VX-445 recently has been proposed to include NBD1 and TMD1 [20] and was determined by cryo-EM to consist of residues in TMD1 and TMD2, leading to stabilization of NBD1 [50].

To narrow down the domain(s) of CFTR involved in the action of VX-445, we tested the effect of VX-445 on the intracellular stability of full-length CFTR, individual domains, and of truncated CFTR F508del constructs: NBD1, TMD1, TMD1-NBD1 (674X), TMD2, NBD2, and two CFTR constructs lacking NBD2 (1202X and 1219X; Figure 8 and Appendix A). Single-domain and multi-domain CFTR constructs were radiolabeled for 5 and 15 min, respectively, and chased for 1, 2, or 4 h. Detergent lysates were immunoprecipitated with the appropriate anti-CFTR antibody, as indicated in the legend. VX-445 (3 μM) was kept present throughout the experiment and cycloheximide (1 mM) was added during the chase to stop elongation of nascent chains and synchronize the full-length population.

VX-445 did not affect intracellular stability of single CFTR domains, including wild-type NBD1 (Appendix A). In contrast, intracellular stability of CFTR F508del improved in the presence of VX-445, and ER-to-Golgi transport was rescued significantly (Figure 8C and Appendix A); even wild-type CFTR was stabilized by this corrector (Figure 8B and Appendix A). VX-445 clearly needs more than a single domain for its efficacy, and the mode of action, therefore, probably affects domain interfaces.

To determine the minimal construct required for VX-445 function, we determined the effect of VX-445 on truncated CFTR F508del constructs. C-terminally truncated, NBD2-lacking CFTR, 1219X, leaves the ER, but has impaired domain assembly [34]. In contrast, the slightly shorter 1202X is incapable of leaving the ER and is degraded instead [34]. When these constructs contained the F508del deletion, VX-445 improved their intracellular stability and for 1219X also transport to the Golgi (Figure 8D,E and Appendix A). These results imply that VX-445 did not require the presence of the NBD2 domain.

Surprisingly, neither the intracellular stability of CFTR 674X F508del (Figure 8F and Appendix A), nor that of NBD1 F508del (Figure 8G and Appendix A) were altered by VX-445. The binding site of VX-445 is located in the TMD1 and TMD2 domains [50], meaning that VX-445 must require inter-domain interactions to act, for which TMD1-NBD1 (674X) alone was not sufficient. TMD1 and NBD1 indeed interact through ICL1 already early and in absence of downstream domains [25], but increased protease protection of TMD1 requires TMD2 (and NBD2) [18]. VX-445, thus, may well act only on domain assembly of CFTR at a later stage of folding, as VX-770 does.

Taken together, VX-445 requires the minimal construct of TMD1-NBD1-R-TMD2 to exert its activity. This is consistent with VX-445 binding to domain interfaces [46,50]. We therefore would expect the compound to have beneficial effects on the domain assembly of the hypo-responder mutants, and perhaps repair impaired domain assembly.

### 2.13. VX-445 Improves Domain Assembly of Hypo-Responder Mutants

To identify the effect of VX-445 on hypo-responder mutants, HEK293T cells expressing wild-type, A46D, G85E, R560S, G628R, and L1335P CFTR were radiolabeled for 15 min and chased for 0 or 2 h in the presence or absence of VX-809, VX-445, or their combination. VX-445 did not affect total levels of CFTR nor domain folding directly after radiolabeling (Figure 9A). Similarly, after 2 h of chase, VX-445 barely affected wild-type, R560S, and G628R CFTR, with again no detectable effect on G85E. VX-445 alone did improve total levels of the A46D mutant and its transport to the Golgi complex (Figure 9 and Figure 10). Domain assembly followed this same pattern, probably because only assembled CFTR can leave the ER: VX-445 strongly increased domain assembly of only the A46D mutant, visible by the massive increase in (upshifted) ‘T1d-f’, T2c, and N2a fragment intensities (Figure 9B and Figure 10).

G85E and L1335P were not responsive to VX-445. Neither domain folding, domain assembly, ER-to-Golgi transport, nor total amount of CFTR were affected. VX-445 clearly does not correct the primary folding defects in these mutants, nor can it rescue these mutants through improved domain assembly.

Adding VX-809 to VX-445 showed strong, additive effects on the A46D, R560S, and G628R mutants. Domain assembly increased, as did export from the ER to the Golgi, as did the total amount of CFTR in the cell (Figure 9B and Figure 10). The strong combined effect of VX-809 and VX-445 on these three hypo-responders implies that VX-809 does contribute to rescue, even when functional responses are undetectable. Every CFTR mutant may benefit from a specific combination of different corrector compounds for their functional improvement and type-I correctors should be considered a component.

## 3. Discussion

In general, CF-causing mutations decrease the number of CFTR molecules at the plasma membrane (N) and/or functionality of the individual CFTR ion channels (the product of Po, open probability, and i, the current that flows through each channel). When mutant CFTR is produced, for example with missense mutations, the reduced net function is the sum of many steps in CFTR biosynthesis: core glycosylation in the ER (band B), domain folding, domain assembly, exit from the ER, transport to the Golgi, glycan modifications in the Golgi (band C), transport to the plasma membrane, and—controlling cell-surface stability—endocytosis and recycling at the plasma membrane. What then does it mean, in terms of these many steps, when a CFTR mutant is a hypo- or hyper-responder? This study set out to redefine the phenotype of hypo- and hyper-responding CFTR mutants to clinical modulators. The aim was to enhance understanding of their defects, of the requirements to fold CFTR into a functional protein, and through this determine the modulator mode of action and mutant(-class) theratyping.

We used the G85E mutation as prototype hypo-responder as it does not respond biochemically nor cell-biologically to the correctors in Trikafta [25,26,27,28,29,48]. Its misalignment of G85E TM1 with the other transmembrane helices in TMD1 [36,37] apparently precludes a straightforward fix. Interesting, then, is that G85E was FDA-approved for Trikafta, based on an >10% functional response detected in FRT cells [21], and that it is reported to be improved also in nasal epithelial cells [16,49]. We did not confirm this with a conformational response. Genetic background may well cause the difference because G85E biosynthesis is improved by modifications of local translation rates through ribosomal-protein Rpl12 depletion [26]. In our studies, a single laboratory cell line is used, allowing comparison of all mutants and wild-type CFTR without a difference in intra- and extra-genic modifiers.

The hypo-responder mutants A46D, G85E, R560S, G628R, and L1335P all are minimal-function proteins. Of these, A46D, G85E, R560S, and G628R were identified as class 2, showing CFTR processing defects [51,52]. Our results confirmed that A46D, G85E, R560S, and G628R are class-2 mutations with strongly decreased CFTR trafficking and cell-surface expression, initiated by misfolding of the domain that carries the mutation. L1335P was identified as a mutant with defective function [53,54,55], and we found it to have a biosynthesis and trafficking defect as well, placing it in class 2 next to perhaps classes 3 or 4. This is consistent with L1335P responding to ivacaftor and lumacaftor alone [55]. L1335P processing from core-glycosylated species to a complex-glycosylated protein was decreased and less protein made it to the cell surface. Our conformational assays established that A46D, G85E, and L1335P deviated from wild type and the other mutants, which may well imply their classification not only in class 2 but also in classes 3 or 4. These unusual conformations were not normalized to wild type with the two corrector drugs, contributing likely to their low responses. The L1335P functional defect hence may well arise from the combination of lower channel numbers on the cell surface and a conformational defect, leading to both ER and Golgi-modified forms (B and C-band, respectively) reaching the cell surface. This is consistent with its functional response to a corrector and to a potentiator without corrector [55].

The existing classifications did not predict the responses of the mutants. We found that only G85E was a cell-biological hypo-responder, as all other mutants responded well to the modulators. This is no surprise when considering that every single mutant protein that is produced by the cell but is not functional must have a conformational defect. These defects lead to classification based on the impact of the cellular chaperone machinery (also called the quality control or proteostasis system). Molecular chaperones triage proteins for either repair and rescue or degradation, determining protein quantity. This same system supports the folding of proteins and also triages for export from the ER. Once at the cell surface, CFTR still (or again) associates with chaperones [56,57], which likely keep supporting and triaging there as well. A dysfunctional CFTR at the cell surface (classes 3 or 4) must have a conformational defect that has escaped the triage system because the chaperones did not detect anything in the protein with which to associate. Some L1335P, for example, escapes the ER to the Golgi complex but it has not assembled all its domains. Consistent with this, we even found the functional hypo-responder A46D to be cell-biologically hyper-responsive to VX-809 in the steady-state. Better yet, steady-state levels were often higher than the kinetic exit rates would predict. This implies that chaperone assistance at the cell surface stabilizes a mutant CFTR protein that the ER-associated quality-control system (including the same core-chaperone set) barely let pass.

As functional responses measure the number of molecules (N) times the functionality of the channel, the outcome of this product would be close to zero when only one of these parameters is close to zero, while the other parameter may be better than wild-type CFTR. This information is essential for determining which modulators a mutant CFTR requires for functional rescue. Further subclassification of mutants seems futile, as the vast majority of CF-causing mutants respond well to modulators, and the exceptions need detailed analysis to determine their requirements.

Residual CFTR function and drug response have been identified to correlate strongly in vitro and in vivo, as well as in the clinic, regardless of the underlying molecular defect in CFTR [55]. In general, CFTR mutations in classes 1 and 2 have less residual function compared to classes 4, 5, and 6. All tested mutants were minimal-function mutations with low residual CFTR activity of <10% of wild-type CFTR. The class-2 hypo-responder mutations A46D, G85E, and R560S have ≤1% residual CFTR function, and the mixed-class mutation L1335P ~2% [51,55]. The hyper-responder mutations either belonged to class-2 mutations with minimal function (F508del, L206W, and I336K), or class-3/4 mutations with defective channel gating or ion conductance (G461R [22], G1249R [46], and S1251N [44]). L206W and I336K have ~7% residual CFTR function [51,52,55]. The residual function of these hyper-responding mutants is at least seven times higher than the hypo-responder mutants A46D, G85E, and R560S, which belong to the same mutation class. On top of this, I336K improved twice as much as the average mutant in trafficking to the Golgi complex, and overall stability (steady-state levels) increased well above average for four out of the five hyper-responder mutants. These mutants, hence, indeed [55] were hyper-responders at all levels, cell biological, functional, and clinical, on top of the response predicted from residual function.

The two corrector drugs did not rescue the primary defect in the mutant domains of the hypo-responder mutants in our assays, but instead improved domain assembly, which likely explains the generality of the response, its mutation/defect independence. All but a few class-3 and -4 mutants, then, would likely benefit from correctors, and all but a few class-2 mutants, then, would benefit from potentiators. This is underscored by the large number of genotypes (>90% of CF patients) that have been FDA-approved for Trikafta, the combination therapy of VX-770, VX-445, and VX-661 (a VX-809 analog), and by our studies showing 9 out of 10 mutants responding to the modulators, with VX-445 improving even wild type.

Trikafta is now used in the clinic to treat all CF patients heterozygous for F508del; patients with a residual-function mutation in the second allele seem to benefit the most [58]. All five hypo-responsive mutations tested in this study are eligible for Trikafta. We found that the functionally hypo-responding mutants A46D and R560S responded well to VX-445 treatment and that A46D showed a hyper-response to both VX-809 and VX-445. In particular, the combination of VX-809 with VX-445 showed additive effects on CFTR domain assembly, ER-to-Golgi transport, and total CFTR levels, showing the necessity to combine modulators that have limited effect by themselves and demonstrating the relevance of the two-fold biochemical response to VX-809.

VX-445 alone is a stronger corrector than VX-809 alone. We concluded that VX-445 supports domain assembly rather than individual CFTR domains, which is confirmed by modeling [20] and cryo-EM structure [50]. TMD1-NBD1-R-TMD2 was required for its efficacy, suggesting that VX-445 needs—and probably enhances—the assembly of TMD1 and TMD2 with each other and with NBD1, beyond assembly of only the N-terminal half. The cryo-EM structure also points to these domains, showing that VX-445 binds residues in Lasso helix 1 (Figure 2C) and TM2 of TMD1, TM10-11 of TMD2, and stabilizes NBD1 [50]. VX-445 works synergistically with type-I and type-II correctors, and was suggested to have a type-III mode of action, stabilizing NBD1 [49,59]. VX-445 was found to associate with NBD1-F508del (and wild-type NBD1) and partially suppress NBD1 unfolding, degradation, and aggregation in vitro, as shown by surface plasmon resonance experiments [49]. Using different approaches, we however did not recapitulate any change in NBD1 stability, nor NBD1-F508del instability caused by VX-445. The cell also did not detect any VX-445-induced difference in thermodynamic stability of the NBD1 domains we tested, as degradation rates were unaffected. The difference in experimental approaches fully explains this apparent discrepancy. Whereas our assays use protease, and thereby may pull the equilibrium of a weakly folded domain towards unfolded (and digested), conditions used for structure determination may pull the equilibrium towards folded. Indeed, the F508del NBD1 structure closely resembles that of wild-type NBD1 [60] but is much more dynamic [61,62]. Recently, VX-445 was reported to increase ion conductance through CFTR, which would classify it as corrector–potentiator [16,63,64]. The potentiating capacity of VX-445 worked for many CFTR variants across different mutation classes: classes 2, 3, and 4, and is synergistic and additive with VX-770 [64]. Our results indicate that VX-445 does not specifically stabilize the NBD1 domain, but rather acts on domain interfaces, suggesting the addition of a type-IV corrector mechanism, with an indirect, positive effect on NBD1 stability.

The weaker corrector, VX-809, improves TMD1, with improved domain assembly as a secondary effect. For all mutants except G85E an ~two-fold increase in protein expression, for hypo- and hyper-responders alike, with stronger responders in both groups. The two-fold improvement is insufficient on its own to rescue most class-2 mutants due to their low residual activity, but in combination with another corrector the class-I correctors are efficacious.

For effective correction of CF-related CFTR mutants, modulators need to be effective in the (sub)domains that are defective (and not perse mutated). Effects are impossible, of course, when mutations are located in the modulator-binding pocket or in locations involved in the allosteric interactions that are affected by modulators. The VX-809-binding pocket contains residues K68, N71, R74, L195, A198, S364, and K681, and mutations reduce the efficacy of this modulator [17]. Residues 371–380 are also essential: for the positioning of TM6 as VX-809-binding pocket [14,65]. Moreover, removal of lasso helix 2 from the TMD1 N-terminus or mutation of residue W57G in this helix completely disrupts protein expression and abolishes VX-809 sensitivity [38]. Mutating lasso helix 2 probably prevents VX-809-induced conformational rescue, due to structural requirements that are no longer met for VX-809 activity.

Most mutants that respond well to VX-809 are located in the transmembrane domains and the strongest responders are the mutants located in intracellular loops (ICLs) [55]. Previous studies showed that VX-809 stabilizes TMD1 [25], promotes interactions between TMD1 and NBD1 via ICL1, and between TMD2 and NBD1 via ICL4 [15,66], but does not directly affect NBD1, TMD2, and NBD2. We here confirmed that the TMD mutants responded strongest to VX-809 but the same was true for the response to VX-445, which has a different mode of action. Docking and molecular-dynamics simulations suggest that VX-445 also targets the TMD1-NBD1 axis [20], to which we experimentally have added a role for TMD2. Indeed, VX-445 stabilizes several TMD1-TMD2 interfaces including residues in Lasso helix 1 [50]. This may explain the hyper-response of A46D to VX-445, as this mutation induces a conformational change that protects Leu49, affecting the N-terminus, also likely the upstream parts targeted by VX-445.

L1335P CFTR is an interesting mutant, which was rescued to the cell surface by VX-809, but barely displayed a response in the pulse-chase experiment, neither to VX-809 nor VX-445. Some export from the ER was allowed in the presence of modulators, while domain assembly was minor and only detected for TMD1 and NBD1. Some small conformational changes appeared in TMD2 (some T2c and loss of T2b) and NBD2 (a weak fragment smaller than N2a), which suggests that these domains are changed during the chase. A clear sign of domain assembly and stabilization of TMD2 and NBD2 was lacking, though. Modest changes in local structure can already prohibit TMD-NBD interactions [67]. This clearly defective domain assembly must have been sufficient to bury chaperone-binding sites and allow some export from the ER, bypassing the Golgi-resident complex-glycosylation machinery or bypassing the Golgi altogether.

Correction and rescue of CFTR mutants requires an increase in N or an increase in functionality. Strong correction by modulators may be sufficient already, but, in general, mutants benefit from (or require) a potentiator. Potentiators such as VX-770 increase the opening rate and the open time of CFTR, increasing the flow of ions through the activated channel [68]. VX-770, therefore, can only improve chloride transport when CFTR is expressed on the cell surface [9]. Class-2 mutants have strongly decreased ER exit and cell-surface CFTR, and therefore will not respond functionally to VX-770. We indeed did not see a conformational effect of VX-770 until CFTR (mutants) had assembled their domains, which took more than 15–30 min in the majority of CFTR molecules.

Too large a number of CF patients with missense mutations do not yet benefit from modulator treatment. The underlying cause of the variability in modulator response is unknown and we are still unable to predict the therapeutic outcome of modulator treatment. The cell-biological/biochemical studies of CFTR mutants combined with an increased knowledge on the mode of action of CFTR modulators will help establish the efficacious modulator combination for each genotype. This study shows that most functionally hypo-responsive mutants respond well to modulator treatment at the conformational and cell-biological level. These modulators, then, will contribute to the functional improvement of CFTR when combined with corrector compounds of a different class. Although the number of hypo-responder missense mutations studied here encompass a small fraction of all missense variants, they are very different, with mutations in different domains, very different defects, and still with good responses. We, therefore, anticipate most functional hypo-responding mutants to be sensitive to modulators biochemically and to improve functionally when combined with another modulator. Individual studies of the exceptionally responding mutants will be needed, as the present classification system does not suffice. Even the classification into functional hypo-responders versus hyper-responders, while matching with clinical phenotypes, does not do justice to the potential of each mutant to function.

## 4. Materials and Methods

### 4.1. Antibodies and Reagents

The generation of polyclonal rabbit antisera MrPink, E1-22, and TMD2C, raised against NBD1, TMD1, and TMD2, respectively, was described previously [18,24,30]. The monoclonal mouse antibody 596 against NBD2 epitope 1204-1211 was kindly provided by Dr. John Riordan (University of North Carolina, Chapel Hill, NC, USA). CFTR modulators VX-809, VX-770, and VX-445 (Selleck Chemicals, Houston, TX, USA) were dissolved in DMSO and stored at −80 °C.

### 4.2. Functional Assays

Patient-derived intestinal organoids were used to determine CFTR function in a well-established assay [22,30,45,46,69]. Forskolin-induced organoid swelling (FIS)—at the indicated forskolin concentrations—is measured, quantified as AUC (area under the curve), and corrected for DMSO-induced swelling. A panel of 10 genotypes was used and their responses to VX-809/VX-770 was determined. The plot in Appendix A shows a large dataset from [9], which shows the corrected function over the uncorrected function, and the correlation using a linear regression model. As the R^2^ in the correlation plot reached 0.74 with high significance, this plot allows the comparison between residual function and drug response. Our mutant selection then was plotted in Appendix A, which shows the hypo-responders to respond worse than most F508del mutants, and the hyper-responders to respond better than the best F508del responders. We included G628R because of the difference between the three patient samples, and added mutant R560S, a previously reported hypo-responder [70].

### 4.3. Expression Constructs

Wild-type CFTR and mutants F508del, G85E, L206W, I336K, and S1251N in pBI CMV2 were a kind gift of Dr. Phil Thomas (UT Southwestern Medical Center, Dallas, TX, USA). The PCR primers that were used to generate the A46D, G461R, R560S, G628R, G1249R, and L1335P point mutants are listed in Appendix A. C-terminal 395X, 674X, 1202X, and 1219X truncation constructs were generated before in pBI CMV2 and pcDNA3 [25,31]. Construct nomenclature: C-terminal truncation Y1219X has a stop codon instead of Y at position 1219, and therefore has 1218 as the most C-terminal residue. pcDNA3 NBD1-CFTR (cDNA 389–673) and pBI CMV2 TMD2-3HA were described before [18,24]. NBD2-CFTR (1202–1480) was generated by PCR with hCFTR as template. The PCR products were cloned into pBI CMV2 using NotI and SalI. F508 was deleted in the truncated constructs by site-directed mutagenesis. All constructs were sequence verified.

### 4.4. Cell Culture and Transient Expression

HEK293T cells were maintained in DMEM supplemented with 10% fetal bovine serum (FBS) and 2 mM Glutamax. All reagents were purchased from Gibco. Cells were cultured in humidified incubators at 37 °C in 5% CO_2_. HEK293T cells were cultured on poly-L-Lysine (Sigma, St. Louis, MO, USA) coated dishes to improve adherence. Cells were transfected 24 h before the experiment, using the 40 kDa polymer polyethylenimine (PEI; Polysciences). The DNA/PEI mixtures (ratio 1:3 (*w*/*w*) with 12.5 μg PEI for a 6-cm dish) were pre-incubated for 20 min at room temperature in 150 mM NaCl. The formed complexes were added to the cells and the medium was exchanged 4 h later.

### 4.5. Surface Biotinylation

Four hours post-transfection, HEK293T cells received complete medium containing 3 µM VX-809, 3 µM VX-445, or no drug, and were grown for 16 h. The cells then were transferred to ice and washed twice with ice-cold PBS^++^ (137 mM NaCl, 2.7 mM KCl, 8 mM Na_2_HPO_4_, 2 mM KH_2_PO_4_, 1mM CaCl_2_, and 0.5 mM MgCl_2_) and left on ice for 15 min in the presence or absence of 3 µM VX-770. After buffer aspiration, the cells were incubated for 30 min with 0.5 mg/mL Sulpho-NHS-SS-biotin (Thermo Fisher Scientific, Waltham, MA, USA) in PBS^++^. Unbound biotin was washed away with two PBS^++^ washes containing 1% BSA. Cells were lysed with 1% Triton X-100 in MNT (20 mM MES, 100 mM NaCl, 30 mM Tris-HCl pH 7.4) and cell lysates were centrifuged at 15,000× *g* at 4 °C to remove nuclei. A small aliquot of the supernatant was mixed 1:1 with sample buffer to a final concentration of 200 mM Tris-HCl pH 6.8, 3% SDS, 10% glycerol, 0.004% bromophenol blue, 1 mM EDTA, and 25 mM DTT (input material). The remaining lysate was incubated with Neutravidin beads (Thermo Fisher Scientific) for 1 h at 4 °C. The samples then were washed twice in a wash buffer consisting of 10 mM Tris-HCl pH 8.6, 300 mM NaCl, 0.1% SDS, and 0.05% Triton X-100 before the beads were eluted in 10 mM Tris-HCl pH 6.8, 1 mM EDTA. Sample buffer was added to a final concentration of 200 mM Tris-HCl pH 6.8, 3% SDS, 10% glycerol, 0.004%, 1 mM EDTA, and 25 mM DTT.

### 4.6. Immunoblotting

Samples were heated for 5 min at 55 °C before loading onto an SDS-PA gel (7.5% for full-length CFTR). Proteins were transferred to PVDF membrane (Milipore) and the membrane was blocked for 1 h in Intercept PBS blocking buffer (LiCor), diluted 1:1 in PBS. CFTR was detected using monoclonal 596 (1:5000) and actin with polyclonal anti-actin (1:5000, Sigma-Aldrich) in PBS supplemented with 0.1% Tween-20. The membranes were washed four times with PBS containing 0.1% Tween-20 and subsequently incubated for 1 h with goat-anti-mouse Alexa 800 (1:10,000) or donkey-anti-rabbit Alexa 680 (1:10,000) in PBS supplemented with 0.1% Tween-20 and 0.02% SDS. Before imaging, the membranes were washed three times with PBS containing 0.1% Tween-20 and a final wash with PBS. Blots were imaged on LiCoR Odyssey CLx according to the manufacturer’s protocol.

### 4.7. Pulse-Chase Analysis

Transfected HEK293T cells at 70–80% confluency were used for pulse-chase analysis as described previously [25,71]. Cells were starved in DMEM supplemented with unlabeled cysteine and methionine (2 µM each) for 15 min. Cells then were radiolabeled for 15 min with 143 µCi/dish Easytag^TM 35^S Express Protein Labeling Mix (Perkin Elmer, Waltham, MA, USA) and chased for indicated times in DMEM supplemented with 5 mM unlabeled methionine and cysteine. CFTR modulators VX-809 (3 µM), VX-770 (3 µM), and VX-445 (3 µM) were kept present during the starvation, pulse, and chase. Where indicated, 1 mM cycloheximide (Carl Roth, Karlsruhe, Germany) was added during the chase. Cells were transferred to ice and washed twice with HBSS before lysing with 1% Triton X-100 in MNT.

### 4.8. Limited Proteolysis, Immunoprecipitation, and SDS-PAGE

Cell lysates were centrifuged for 10 min at 15,000× *g* at 4 °C to remove nuclei. A fraction of the cell lysate was used immediately to immunoprecipitate CFTR with polyclonal MrPink. The remainder of the lysate was incubated with Proteinase K from Tritirachium album (Sigma) for 15 min. Limited proteolysis was stopped by addition of 2.5 mM PMSF. The (proteolyzed) lysates were transferred to Protein-A Sepharose beads (GE Healthcare, Chicago, IL, USA) that had been pre-incubated for 15 min with antibodies at 4 °C. After overnight immunoprecipitation for MrPink and 3 h incubation for E1-22, TMD2C, and 596, the complexes were washed twice at room temperature for 15 min in the following buffers: MrPink in 10 mM Tris-HCl pH 8.6, 300 mM NaCl, 0.1% SDS, and 0.05% Triton X-100, E1-22 in 10 mM Tris-HCl pH 8.6, 300 mM NaCl, 0.05% SDS, and 0.05% Triton X-100, TMD2C in 50 mM Tris-HCl pH 8.0, 150 mM NaCl, 1 mM EDTA, and 596 in 30 mM Tris-HCl pH 7.5, 20 mM MES,100 mM NaCl, 0.5% TX-100. The washed beads were resuspended in 10 mM Tris-HCl pH 6.8, 1 mM EDTA before sample buffer was added to a final concentration of 200 mM Tris-HCl pH 6.8, 3% SDS, 10% glycerol, 0.004%, 1 mM EDTA, and 25 mM DTT. Samples were heated for 5 min at 55 °C before loading onto an SDS-PA gel (7.5% for full-length CFTR, 12% for proteolyzed CFTR). Gels were dried and exposed to film (Carestream Health, Rochester, NY, USA) or to super-resolution Phosphorimager screens (Fuji Film, Tokyo, Japan).

The limited-proteolysis assays are done in the presence of Triton X-100. Use of (this) detergent is essential because Triton X-100 does not denature/unfold CFTR but does release many associating proteins such as molecular chaperones. We here have aimed to study folding and misfolding of CFTR mutants, without interference of associating proteins.

### 4.9. Quantifications

Radioactive band intensities were quantified by a Typhoon ELA-7000 scanner (GE Healthcare Life Sciences, Chicago, IL, USA) using ImageQuantTL software (GE Healthcare Life Science). Lane profiles were determined using the same software.

### 4.10. Structure Analysis

Images of protein structures were created using ChimeraX1.4.

## Figures and Tables

**Figure 1 ijms-23-15170-f001:**
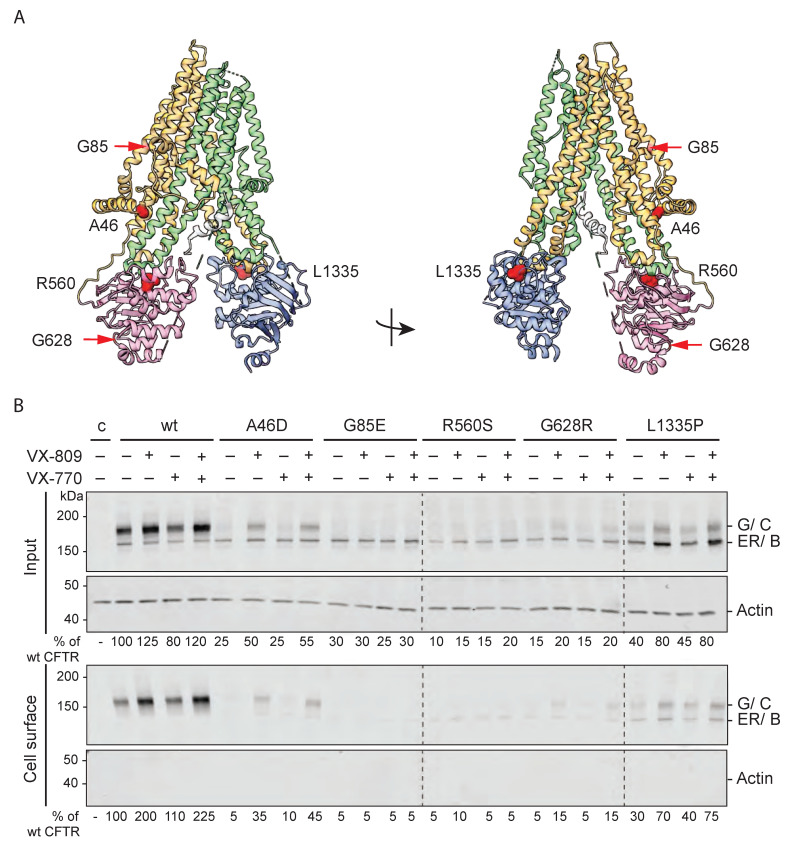
VX-809 improves total CFTR and cell-surface levels of hypo-responding CF-causing mutants. (**A**) Cartoon of CFTR cryo-EM structure (PDB: 5UAK) and location of CF-causing mutations in color-coded CFTR domains (TMD1 in orange, NBD1 in pink, R-region in grey, TMD2 in green, and NBD2 in blue). (**B**) HEK293T cells expressing wild-type CFTR or indicated CFTR mutants were pre-incubated for 18 h with 3 µM VX-809 and/or for 15 min with 3 µM VX-770. Cells then were incubated with 0.5 mg/mL Sulpho-NHS-SS-biotin for 30 min at 4 °C. Unbound biotin was removed by the addition of 1% BSA. Cells were lysed with 1% Triton X-100 and lysates were either subjected to pull-down with Neutravidin beads (cell surface) or used directly for SDS-PA gel analysis (input) on 7.5% SDS-PA gels. CFTR and actin were visualized by Western blotting. Representative blots of three repeats. ER, ER form (also called B band); G, complex-glycosylated CFTR form, which has left the ER and may reside in or beyond the Golgi complex (also called C band); c (control), transfection with empty vector; wt, wild-type CFTR. Quantifications are below each lane and used for plots below.

**Figure 2 ijms-23-15170-f002:**
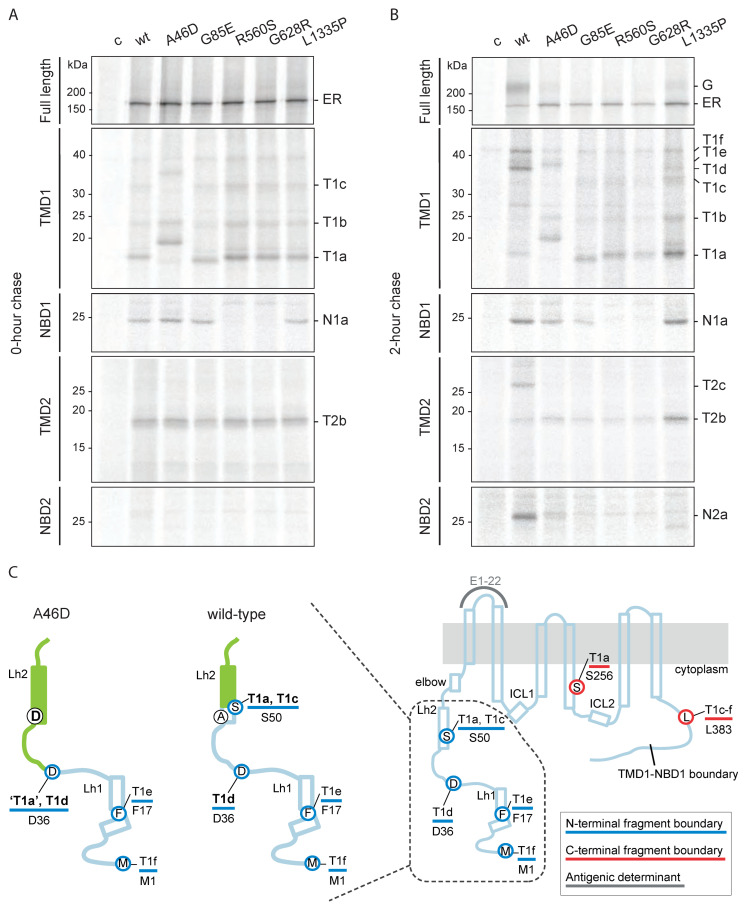
Hypo-responder mutants show defects in molecular folding (**A**) HEK293T-cells expressing wild-type or mutant CFTR were radiolabeled for 15 min. Cells were lysed with 1% Triton X-100 and the detergent cell lysates were subjected to 0 (top panel) or 25 µg/mL proteinase-K digestion for 15 min. Non-proteolyzed CFTR was immunoprecipitated using MrPink (top panel) and domain-specific fragments (lower panels) with E1-22 (TMD1), MrPink (NBD1), TMD2C (TMD2), or 596 (NBD2). Undigested CFTR was analyzed on 7.5% SDS-PA gel and digested CFTR on 12% SDS-PA gel. Representative gel of at least three repeats. Lane profiles of wild-type and G85E T1a-c are shown in Appendix A. (**B**) As in (**A**), but radiolabeled CFTR was chased for 2 h. T1a–f, protease-resistant TMD1-specific fragments; N1a, protease-resistant NBD1-specific fragment; T2a–c, protease-resistant TMD2-specific fragments; N2a, protease-resistant NBD2-specific fragment. (**C**) Cartoon highlighting the A46D-induced changes in the N-termini of proteolytic TMD1 fragments. Wild-type TMD1 fragments are T1a–f. The A46D-induced aberrant fragments result from protection of the S49 region and N-terminal extension of TMD1 fragments T1a and T1c (green-filled): T1a becomes a shifted, larger ‘T1a’, and T1c turns into T1d. Alpha helices are depicted as rectangles; grey line, antigenic epitope E1-22; blue lines and blue-circled residues, N-terminal boundaries of fragments; red lines and red-circled residues, C-terminal boundaries of fragments; Lh1, lasso helix 1; Lh2, lasso helix 2; elbow, N-terminal elbow helix.

**Figure 3 ijms-23-15170-f003:**
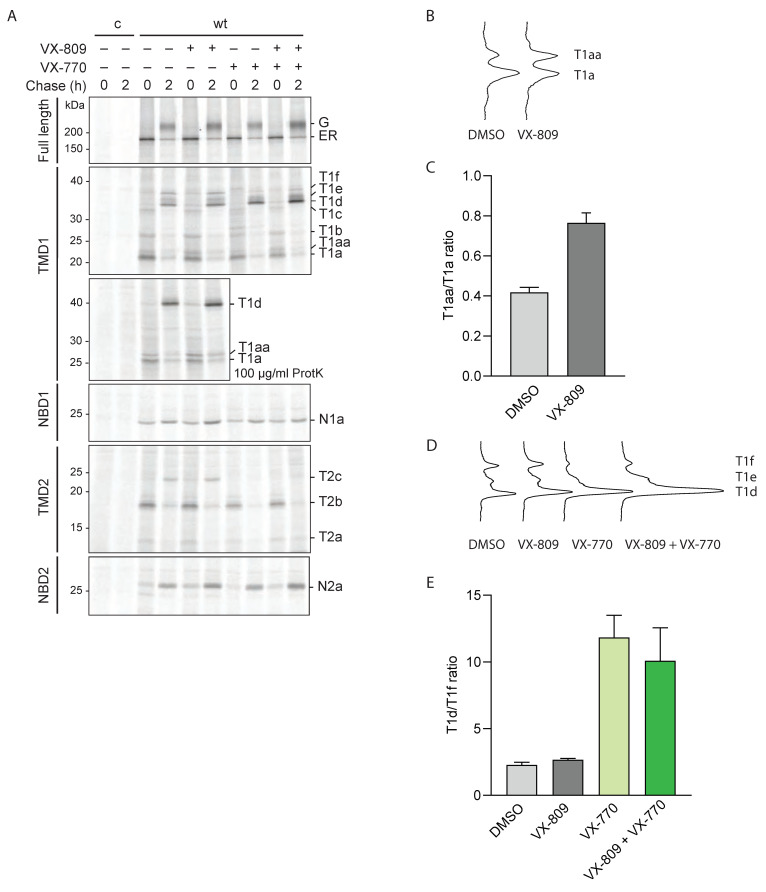
Conformation of CFTR is sensitive to the binding of modulators (**A**) HEK293T cells expressing wild-type CFTR were radiolabeled for 15 min and chased for 0 or 2 h, as in Figure 2. Where indicated, VX-809 (3 µM) or VX-770 (3 µM) were added during starvation, pulse, and chase. CFTR was analyzed as in Figure 2. The top panel shows undigested CFTR and the lower panels show domain-specific fragments immunoprecipitated after digestion with 25 µg/mL or (where indicated) 100 µg/mL proteinase K. Representative gel of three experiments. (**B**) Lane profiles of immunoprecipitated TMD1 fragments T1a and T1aa, after 0 h chase and 100 µg/mL proteinase-K digestion. (**C**) Quantification of T1a and T1aa fragment ratios at 0 h chase. Band intensities were quantified using ImageQuant. (**D**) Lane profiles of immunoprecipitated TMD1 fragments T1d, T1e, and T1f after 2 h of chase. For an explanation of abbreviations see the legends of Figure 1 and Figure 2. (**E**) Quantification of T1d and T1f fragment profiles as in (**C**). Bars show standard deviations of *n* = 3.

**Figure 4 ijms-23-15170-f004:**
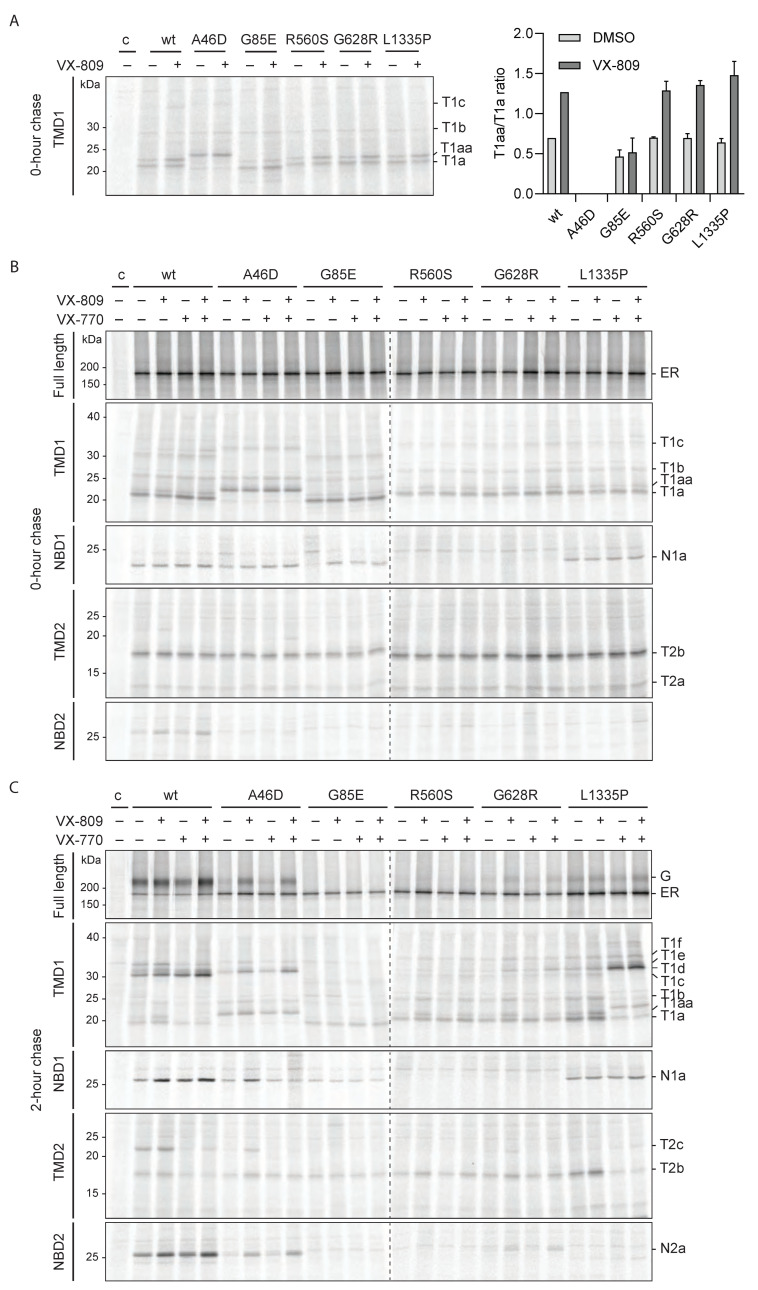
Modulator-specific conformational changes are visible in hypo-responder mutants (**A**) HEK293T cells expressing wild-type and mutant CFTR were radiolabeled for 15 min. Where indicated, VX-809 (3 µM) was added during the starvation, pulse, and chase. Detergent lysates were treated with 100 µg/mL proteinase K and immunoprecipitated using TMD1-specific antibody E1-22. Quantification of TMD1 T1a and T1aa fragments at 0 h chase, as in Figure 3C. (**B**) CFTR was analyzed as in Figure 3A. The top panel shows undigested CFTR and the lower panels show domain-specific fragments immunoprecipitated after digestion with 25 µg/mL proteinase K. (**C**) As in (**A**), but radiolabeled CFTR was chased for 2 h. Gels are representative of three repeats. Lane profiles of wild-type and A46D TMD1 fragments are shown in Appendix A. Lane profiles of wild-type and L1335P NBD2 fragments are shown in Appendix A. Data are used for plots below.

**Figure 5 ijms-23-15170-f005:**
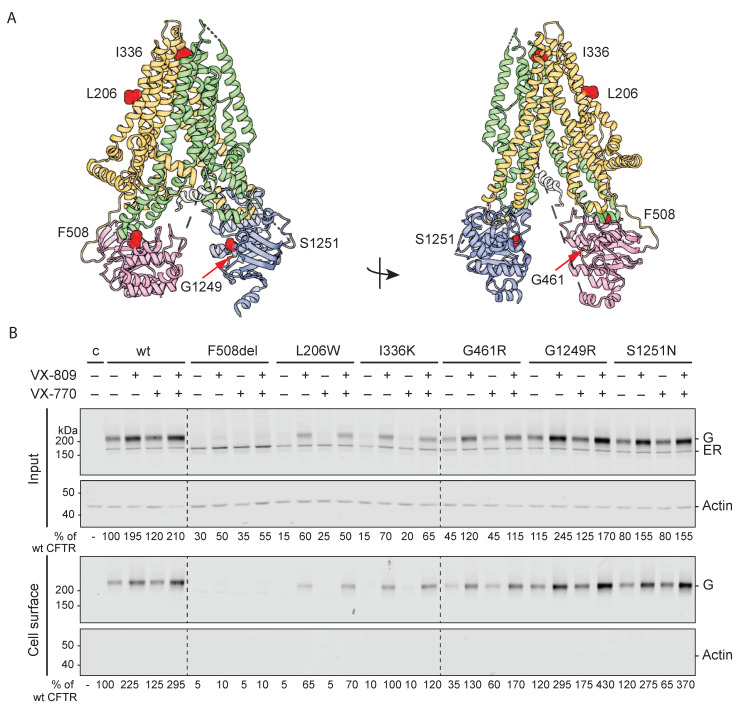
VX-809 strongly improves cell-surface levels of hyper-responding CFTR variants (**A**) Location of CF-causing mutations in the cryo-EM structure of CFTR (PDB: 5UAK) in color-coded CFTR domains (TMD1 in orange, NBD1 in pink, R-region in grey, TMD2 in green, and NBD2 in blue). (**B**) HEK293T cells expressing wild-type CFTR or indicated CFTR mutants were pre-incubated for 18 h with 3 µM VX-809 and/or for 15 min with 3 µM VX-770. CFTR was analyzed as in Figure 1B. Gels are representative of at least three independent experiments. Quantifications are below each lane and used for plots below.

**Figure 6 ijms-23-15170-f006:**
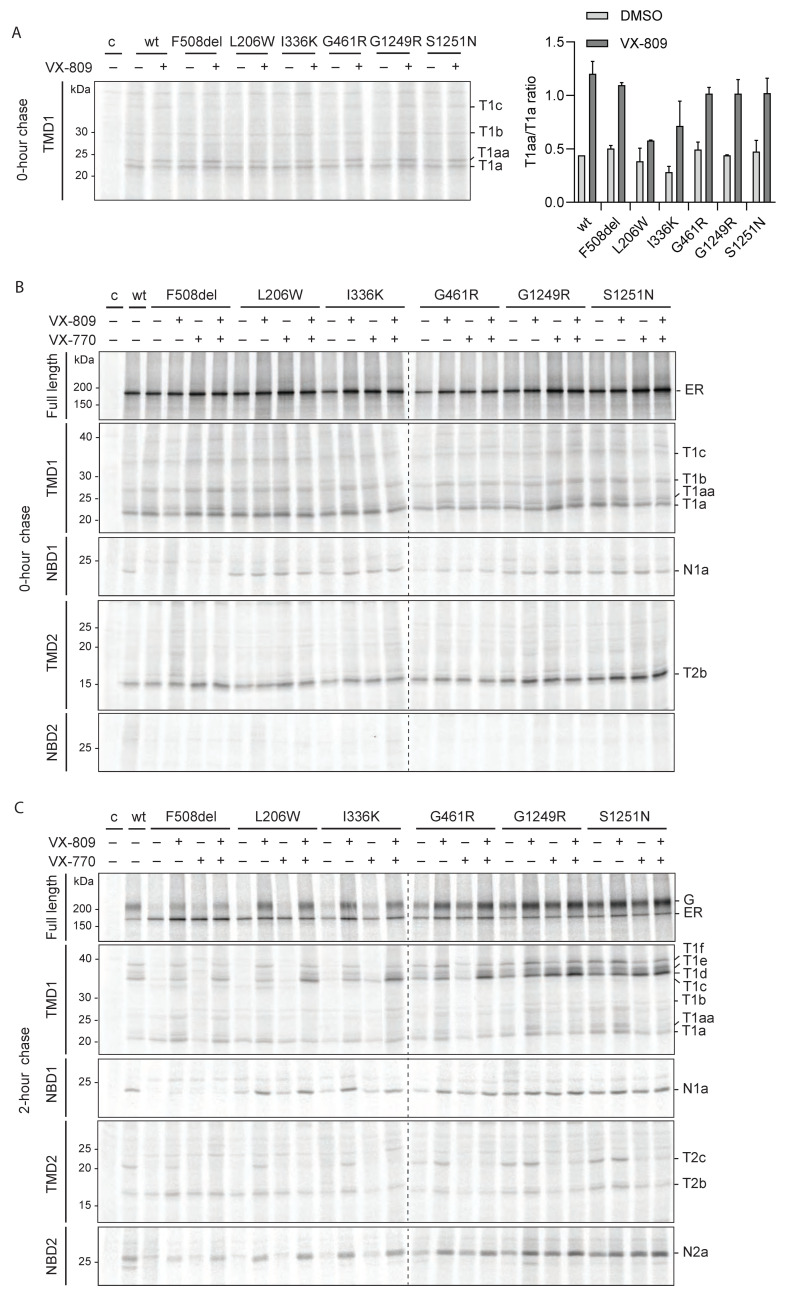
Hyper-responders show conformational responses to modulators (**A**) HEK293T cells expressing wild-type and mutant CFTR were radiolabeled for 15 min. Where indicated, VX-809 (3 µM) was added during the starvation, pulse, and chase. TMD1 proteolysis was analyzed as in Figure 4A. Quantifications based on three independent experiments. (**B**) CFTR was analyzed as in Figure 4B. VX-809 (3 µM) or VX-770 (3 µM) were added during starvation, pulse, and chase. The top panel shows undigested CFTR and the lower panels show domain-specific fragments immunoprecipitated after digestion with 25 µg/mL proteinase K. (**C**) As in (**B**), but radiolabeled CFTR was chased for 2 h. Representative gels for each construct are shown. Data are used for plots below.

**Figure 7 ijms-23-15170-f007:**
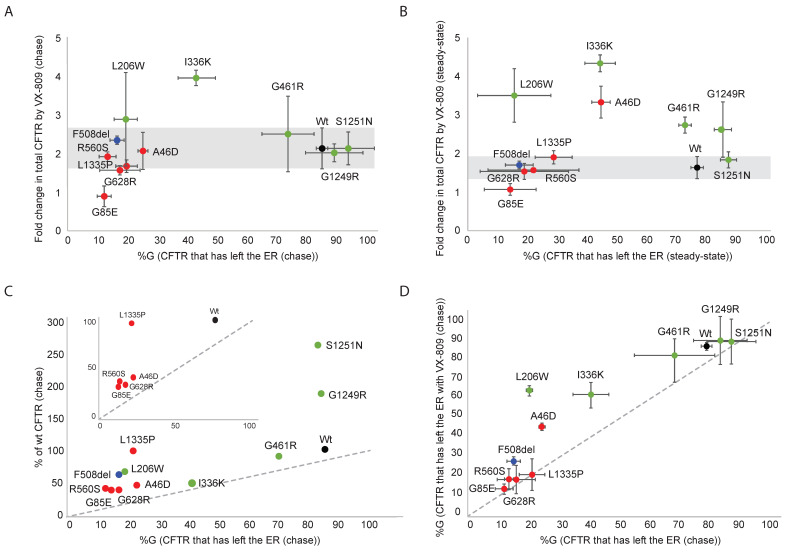
Modulators have comparable effects on hypo- and hyper-responding CFTR mutants (**A**) Fold change in (mutant) CFTR levels (ER+G) by VX-809 (*y*-axis) at 2 h of chase plotted against the percentage CFTR that had left the ER after 2 h of chase (G/(ER+G)) in the DMSO control (*x*-axis) (n ≥3 for each mutant, including data from Figure 4 and Figure 6). The cut-off for a wild-type-like VX-809 response, shown as a light grey bar, is the standard deviation of wild-type CFTR samples (*n* = 6). Hypo-responder mutants are depicted in red and hyper-responders are depicted in green. (**B**) As in (**A**), but in steady-state. Graphs include data from Figure 1 and Figure 5. (**C**) CFTR (ER + G) that is present at 2 h of chase, quantified from Figure 4 and Figure 6, as percentage of wild-type CFTR (*y*-axis), plotted against the percentage of CFTR that had left the ER at that time point (G/(ER + G)), both in DMSO control (*x*-axis). Insert shows the hypo-responder mutants in the same plot with a *y*-axis to 100%. (**D**) The percentage CFTR that had left the ER after 2 h of chase in the presence of VX-809 (*y*-axis) plotted against the percentage CFTR that had left the ER after 2 h in the DMSO control (*x*-axis). Error bars correspond to quantifications of at least three independent experiments, including data from Figure 4 and Figure 6.

**Figure 8 ijms-23-15170-f008:**
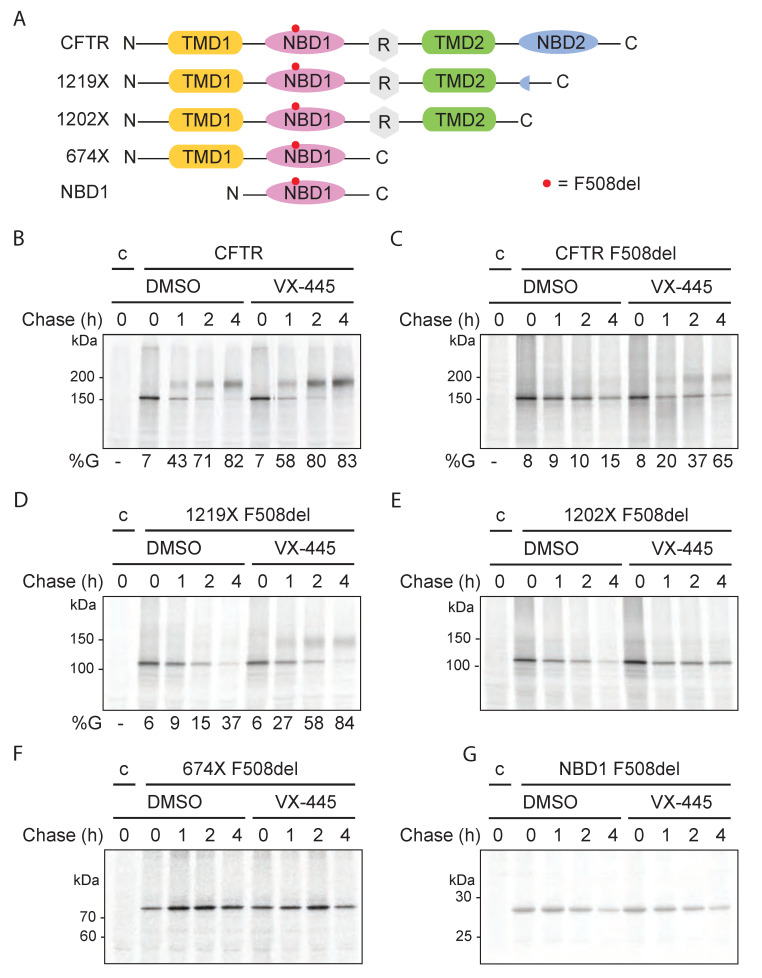
The action of VX-445 requires TMD1-NBD1-R-TMD2 (**A**) Schematic representation of constructs used in (**B**–**G**). (**B**) HEK293T cells expressing full-length CFTR was radiolabeled for 15 min and chased for 0, 1, 2, or 4 h. VX-445 (3 µM) was added during starvation, pulse, and chase and cycloheximide (1 mM) was added during the chase. CFTR was immunoprecipitated from detergent cell lysates using MrPink and analyzed on 7.5% SDS-PA gels. (**C**–**E**) as in (**B**), intracellular stability of (**C**) Full-length F508del CFTR, (**D**) 1219X F508del CFTR, and (**E**) 1202X F508del CFTR. (**F**) As in (**B**) but 674X F508del CFTR was radiolabeled for 5 min and chased for 0, 1, 2, or 4 h. Samples were immunoprecipitated with MrPink and analyzed on 12% SDS-PA gels. (**G**) As in (**B**) for F508del NBD1. Quantifications are in Appendix A.

**Figure 9 ijms-23-15170-f009:**
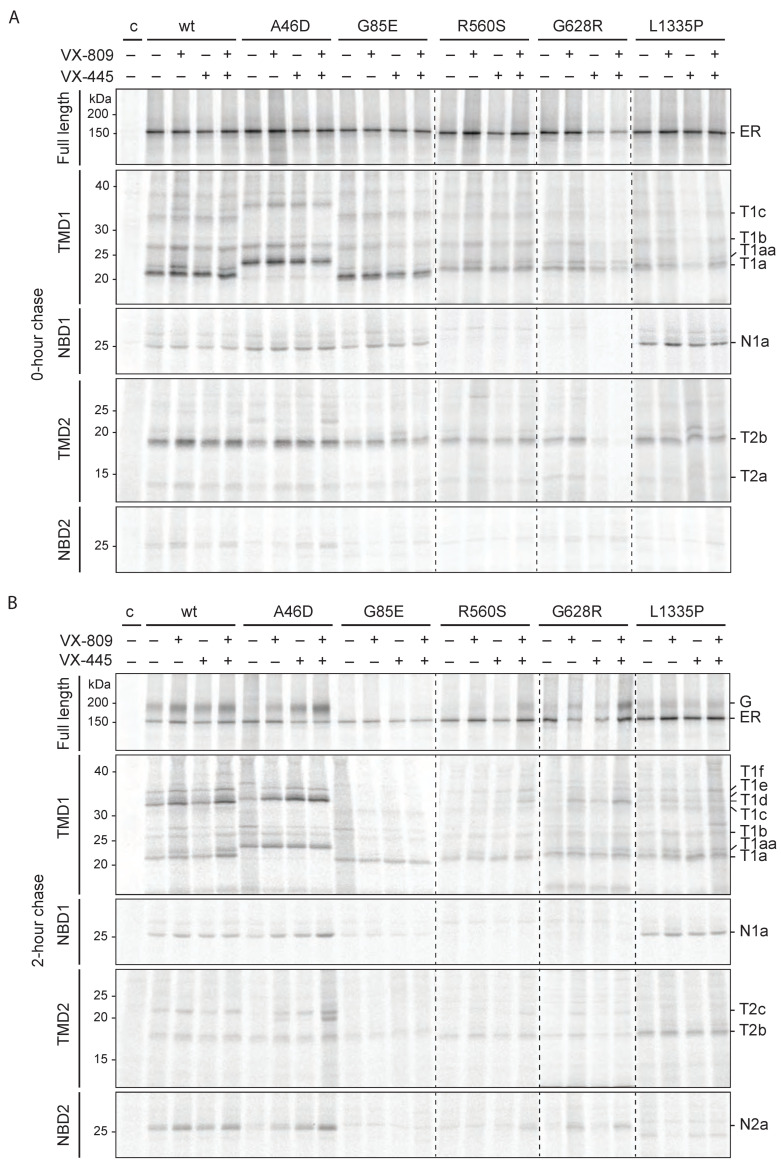
Some hypo-responders are rescued by combination treatment with VX-445 (**A**) HEK293T cells expressing wild-type and mutant CFTR were radiolabeled for 15 min. VX-809 (3 µM) was added during starvation, pulse, and chase. VX-445 was added during the starvation (3 µM), pulse (3 µM), chase (3 µM), stop on ice (30 µM), and lysis on ice (30 µM). CFTR was analyzed as in Figure 4. The top panel shows undigested CFTR and the lower panels show domain-specific fragments immunoprecipitated after digestion with 25 µg/mL proteinase K. (**B**) As in (**A**), but radiolabeled proteins in the cells were chased for 2 h. Quantifications are in Figure 10.

**Figure 10 ijms-23-15170-f010:**
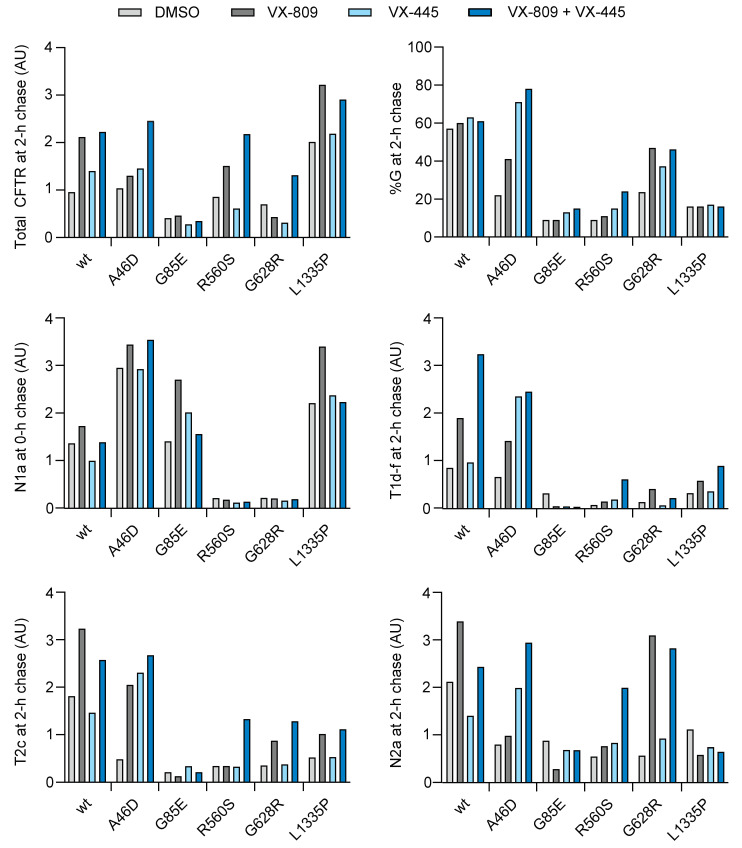
Quantification of hypo-responding CFTR mutants in the presence of modulators. Graphs depicting the total amount of (mutant) CFTR, percentage of CFTR that was transported to the Golgi complex, and the intensity of CFTR fragments in the presence of solvents (DMSO), VX-809, VX-445, or both, from Figure 9.

## Data Availability

All data are provided within the manuscript. Alternative formats are available upon request.

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
