# Peer review of "Redefining Hypo- and Hyper-Responding Phenotypes of CFTR Mutants for Understanding and Therapy"

_ijms, 2022, doi:10.3390/ijms232315170_

Round 1
Reviewer 1 Report
The study by Hillenaar et al. utilizes CFTR mutants that are either hypo- or hyper-responders to first generation CFTR modulators and investigates their biogenesis, cell surface expression and folding.
Major concerns:
1. Large parts of the manuscript are quantitative analysis and discussions on what seems to be single experiments without any statistical evaluation of the results. The experiments used, immunoblotting, cell surface biotinylation followed by immunoprecipitation and immunoblotting, and pulse-labeling in combination with protease susceptibility with or without chase, are prone to experimental variation in the hands of even the best experimental researcher. Thus, multiple experiments with appropriate statistical analysis of the experimental variation are a prerequisite for their quantitative discussion in the manuscript. Also a copy number normalization between different transfections has to be performed in order to compare between mutants.
2. The functional measurements that are the basis of the hypor- and hyperresponder identification are not part of the current manuscript. The authors refer to a preprint manuscript, however in this manuscript only the absolute organoid swelling in presence of VX-770 and VX-770+VX-809 are shown. To identify hypo- or hyperresponders the ratio of the corrected divided by the uncorrected function has to be analyzed, an analysis which has to be available either in this or in the cited manuscript.
3. In the protease susceptibility immunoblots the authors identify a large number of individual fragments and discuss these as if their precise molecular identity was known. If this identification was performed in a previous manuscript, better description of the reference should be provided. Otherwise identification of these bands e.g. by mass-spec has to be performed.
4. The authors suggest that for some mutants, particularly L1335P but also R560S and G628R, core-glycosylated band B reaches the cell surface (Fig 1B). If this was true a widely used assay to determine CFTR correction (C/B+C ratio) would be invalid. Additional experiments have to be performed to confirm this finding. These could include proteasomal inhibition in combination with cell surface biotinylation and differential stability measurement of the B band in the ER and at the cell surface.
Minor points:
- The G85E has been shown to respond to Trikafta correction in several publications (e.g. PMID: 33303536, PMID: 32853178).
- The mutants G461R, G1249R and S1251N are referred to as class-4 mutants. This would mean that their unitary conductance rather than their open-probability is reduced. To the opinion of this reviewer this has never been shown. Also the L1335P mutation is referred to as class-4 mutation with the supporting citation of two clinical papers that do not contain data to distinguish between a gating and conductance defect.
- The paper by Han et al JCI Insight . 2018 Jul 26;3(14):e121159 describes the identification and characterization of hyper-responsive mutants. While this manuscript is cited in another context, comparisons of these results to the findings in the current manuscript are missing.
Reviewer 2 Report
Overall, the proteolysis experiments are commendable for revealing the effect of mutations and small-molecule compounds (modulators) on the CFTR conformation/folding. However, it is a serious issue that CFTR fragment quantification has not been performed properly and statistical analysis has not been performed. The authors should quantify most of the CFTR fragments (e.g., T1a, aa, N1a, T2b, N2a, etc.) from multiple experiments. Most of the data seem to be from only a single experiment which concerned with reproducibility. Moreover, the paper is very confusing and unorganized.
The authors performed the limited proteolysis analysis using the CFTR solubilized with detergent which could disrupt the conformation. As Lukacs lab performed, the authors should perform the limited proteolysis experiments using the CFTR in microsomes prepared without any detergents.
Round 2
Reviewer 1 Report
The manuscript by Hillenaar et al. has been substantially improved during the revision and my queries have been answered. I therefore recommend the manuscript for publication.
Reviewer 2 Report
Thank you for correcting the manuscript. Now I am happy to accept your manuscript for publication in IJMS.